# Glycine: The Smallest Anti-Inflammatory Micronutrient

**DOI:** 10.3390/ijms241411236

**Published:** 2023-07-08

**Authors:** Karla Aidee Aguayo-Cerón, Fausto Sánchez-Muñoz, Rocío Alejandra Gutierrez-Rojas, Lourdes Nallely Acevedo-Villavicencio, Aurora Vanessa Flores-Zarate, Fengyang Huang, Abraham Giacoman-Martinez, Santiago Villafaña, Rodrigo Romero-Nava

**Affiliations:** 1Escuela Superior de Medicina, Instituto Politécnico Nacional, Sección de Estudios de Posgrado e Investigación, Ciudad de Mexico 11340, Mexico; aidee.aguayo@gmail.com (K.A.A.-C.); dranallelyacevedo0611@gmail.com (L.N.A.-V.); vanefloz21@gmail.com (A.V.F.-Z.); santiagovr1@gmail.com (S.V.); 2Departamento de Inmunología, Instituto Nacional de Cardiología “Ignacio Chávez”, Ciudad de Mexico 14080, Mexico; fausto22@yahoo.com; 3Escuela Nacional de Ciencias Biológicas, Instituto Politécnico Nacional, Ciudad de Mexico 11350, Mexico; ross.grojas.22@gmail.com; 4Laboratorio de Investigación en Obesidad y Asma, Hospital Infantil de México Federico Gómez, Ciudad de Mexico 06720, Mexico; huangfengyang@gmail.com; 5Laboratorio de Framacología, Departamaneto de Ciencias de la Salud, DCBS, Universidad Autónoma Mteropolitana-Iztapalapa (UAM-I), Ciudad de Mexico 09340, Mexico; agmfest.be@gmail.com

**Keywords:** glycine, targets, inflammation, immunomodulator

## Abstract

Glycine is a non-essential amino acid with many functions and effects. Glycine can bind to specific receptors and transporters that are expressed in many types of cells throughout an organism to exert its effects. There have been many studies focused on the anti-inflammatory effects of glycine, including its abilities to decrease pro-inflammatory cytokines and the concentration of free fatty acids, to improve the insulin response, and to mediate other changes. However, the mechanism through which glycine acts is not clear. In this review, we emphasize that glycine exerts its anti-inflammatory effects throughout the modulation of the expression of nuclear factor kappa B (NF-κB) in many cells. Although glycine is a non-essential amino acid, we highlight how dietary glycine supplementation is important in avoiding the development of chronic inflammation.

## 1. Introduction

Glycine, also known as amino acetic acid, is an important component of many proteins and plays a crucial role in the synthesis of many biomolecules, including creatine and purine nucleotides [1]. Glycine was first isolated in 1820 from the acid hydrolysis of gelatine. Its name is derived from the Greek word glykys, meaning sweet, and it is the smallest amino acid (with a molecular weight of 75.067 g/mol). It is located in both the hydrophilic and hydrophobic parts of the polypeptide chain [2,3]. It is abundant in plasma and represents 11.5% of the total amino acids and 20% of the nitrogen in body proteins and accounts for 80% of protein [4,5]. The necessary dietary intake of glycine is ~1.5–3 g/day [6] given that in young men, glycine flux is 34–35 mg/kg/h on average in the fed state; during the post-absorptive state, the glycine flux is decreased by half (around 18 mg/kg/h) [7,8]. Around 35% of glycine in the body comes from endogenous synthesis [9], and the average rate of whole-body de novo glycine synthesis is estimated at 12–15 mg/kg/h, contributing to 81% of the systemic flux [2,7,10]. The physiological glycine plasma concentration ranges from 200 to 300 mol/L [11].

Glycine is synthesized endogenously by the body from serine, choline, threonine, and glyoxylate [12,13]; hence, it has been classified as an unessential amino acid for mammals [4] that has many activities in different systems (Figure 1). Glycine acts as a neurotransmitter and modulates neuronal activity [14]; its main activity is related to the inhibition of different brain regions. For example, in the central nervous system (CNS), glycine binds to chloride-sensitive ion channels to inhibit postsynaptic neurons [15]. It plays an important role in the mechanism of pain transmission: pharmacological treatment or genetic deletion that inhibits glycinergic signaling is sufficient to evoke pain hypersensitivity in living organisms [16]. Glycine has also been associated with the control of motor functions due to its ability to ameliorate motor deficiencies after surgery [17]. Glycine also plays an important role in the regulation of gene expression [18], protein configuration and activity, and several other biological functions [19]. There are other beneficial activities in which glycine is involved: as antacid, modulator of growing throughout the regulation of growth hormone (GH) synthesis; improves muscle tone, collagen synthesis, tissue restore (scar formation) and delaying muscular degeneration [20], in addition, it has been reported that glycine also protects the intestine against the harmful effects of radiotherapy in cancer treatment [21].

Glycine plays a role in diabetes. It is a secretagogue of glucagon-like peptide-1 (GLP-1) [22], insulin, and glucagon [23] because it has been shown that the effect of ingested glycine on the postprandial glucose concentration facilitates the secretion of insulin by other amino acids [24].

Decreased glycine receptor (GlyR) expression in cells from people with type 2 diabetes mellitus (T2DM) is associated with a disruption of glycine-induced insulin secretion [25]. Clinical studies have shown that higher circulating glycine concentrations help lower the risk of developing T2DM [26].

The objective of this review is to integrate information from basic and clinical studies regarding the role of glycine as a therapeutic agent to regulate the low-grade inflammation associated with disease. We also integrated the possible mechanism throughout glycine could act as a ligand and it has an effect by activation of different pathways related to inflammation process in a variety of cells that belong to different systems.

## 2. Glycine Targets

According to the International Union of Basic and Clinical Pharmacology (IUPHAR), glycine has the following natural/endogenous targets: GlyRs (with α1, α2, α3, α4, and β subunits), a co-agonist of ionotropic glutamate receptors (GluN1, GluN2A, GluN2B, GluN2C, and GluN2D), G protein-coupled receptor family C group 6 (GPRC6), and transporters, which move this compound across lipid membranes. The transporters include glycine transporter type 1 and 2 (GlyT1 and GlyT2, respectively), proton-coupled amino acid transporter 1, vesicular inhibitory amino acid transporter, proton-coupled amino acid transporter 2, neutral amino acid transport (B0AT1, B0AT2, B0AT3, and NTT4), sodium-coupled neutral amino acid transporter 1, sodium-coupled neutral amino acid transporter 2, sodium-coupled neutral amino acid transporter 4, and sodium-coupled neutral amino acid transporter 5 [27,28,29,30].

## 3. Receptors

### 3.1. GlyRs

GlyRs are ligand-activated pentameric ion channels that belong to the Cys-loop family of transmitter-activated ion channels (zinc-activated channels). This family also include γ-aminobutyric acid receptor type A (GABAA); nicotinic acetylcholine receptors; N-methyl-D-aspartate (NMDA) receptors, which are ionotropic glutamate receptors (iGluRs); and serotonin receptor 5-hydroxytryptamine type 3 (5HT3R) [15,31,32]. GlyRs are abundantly expressed throughout the CNS: there are postsynaptic, presynaptic [15], and extrasynaptic GlyRs [33]. There are four known GlyR α subunits (α1–α4) and a single β subunit in vertebrates [34]. Normally, the β subunit is part of a heteromultimeric complex with GlyR α subunits [35]. 

GlyRs are expressed as homopentamers of five α subunits or as heteropentamers of three α and two β subunits or two α subunits and three β subunits. The receptor is an intrinsic anion channel [34,36,37,38]. As mentioned above, there are four α subunits, namely, α1, α2, α3, and α4, and a single gene coding for the β subunit [39,40]. The four α subunits generally share >90% amino acid sequence homology with each other, but their genes are expressed in specific zones and are developmentally regulated. The α2 subunit is highly expressed in all layers of the cerebral cortex, brain stem, thalamus, spinal cord, hippocampus, diencephalon, and cerebellum during embryonic development [41,42]. There is a change from α2 homomeric GlyRs to α1β heteromeric GlyRs during development [15]. In neonatal period, levels of α1 and α3 expression increase. α1 is prominently expressed in the hypothalamus, colliculi, the spinal cord, and brain stem cerebellar deep nuclei [43,44]. α3 has a relatively lower level of expression than α1 at all developmental stages [45,46]. α4 is an embryonic GlyR subunit isoform [40] but is presumed to be a pseudogene in humans due to a premature stop codon upstream of the final TM4 domain. The β subunit gene is transcribed at all developmental stages and is widely and abundantly distributed in the spinal cord and brain [47,48]. Its distribution is broader than that of α1. The β subunit is indispensable for synaptic clustering [49]. 

When glycine binds to GlyRs, the channel opens (it is formed by the domine 2 of each subunit of the GlyR) and generates a short-term flux of negative ions into the cell. In this way, the intracellular chlorine concentration increases temporarily, leading to hyperpolarization of the membrane, which prevents the cell from being easily excited (Figure 2). The net effect is inhibitory [50,51]. GlyRs are involved in several processes, including the central regulation of orexigenic signals in obesity [52].

### 3.2. NMDA Receptor

The N-methyl-D-aspartate (NMDA) receptors are iGluRs. They share common characteristics with the other two glutamate receptors, namely, α-amino-3-hydroxy-5-methyl-4-isoxazolepropionic acid (AMPA) and 2-carboxy-3-carboxymethyl-4-isopropenylpyrrolidine (kainate) receptors. NMDA receptors are heterotetramers composed of two GluN1 subunits and two GluN2 and/or GluN3 subunits [53,54]. Currently, eight splice variants of the GluN1 gene, four GluN2 genes (GluN2A–GluN2AD), and two GluN3 genes (GluN3A and GluN3B) have been identified [54]. NMDA receptors containing the NR2A, NR2B, and NR2D. These subunits are commonly expressed on spinal dorsal horn neurons. In contrast, there are lower levels of expression of NMDA receptors containing the NR2C subunit in this site [55]. These receptors require the concomitant binding of glutamate and glycine or D-serine as a co-agonist [56]. Glycine has an excitatory effect on these receptors [57] (Figure 2).

### 3.3. GPRC6 Receptor

GPRC6 is a 362 amino acid [58] orphan receptor coupled to a stimulatory Gα subunit (Gαs). It has been associated with the cannabinoid family, class A (similar to rhodopsin) [58,59,60], which phylogenetically belongs to the melanocortin/endothelial differentia-tion/cannabinoid/adenosine (MECA) [61]. It has been shown in oocytes that GPR6 can be activated by glycine, this being a small neutral aliphatic L-α-amino acid that could act as an endogenous agonist in concentrations of 100 µM [62]. GPR6 was previously known as the sphingosine 1-phosphate receptor [63], with sphingosylphosphorylcholine (SPC) considered to be an endogenous ligand [64]. However, it is still considered an orphan receptor [65]. This receptor has a high constitutive activation of adenylyl cyclase—leading to the production of cyclic adenosine monophosphate (cAMP)—and competes with other Gαs-coupled receptors for their corresponding agonists [59,63,66] (Figure 2). In humans, GPRC6 is highly expressed in the hypothalamus [67], leucocytes, skeletal muscles, and testes. In adults, it has medium levels of expression in the heart, kidney, liver, and spleen, and low levels of expression in the pancreas, placenta, ovary, and lung [68]. In previous studies, we found that in adipocytes, glycine has an effect on the expression of GPRC6 [69], and even this receptor has been related to neuroprotective activity [70]; however, there is not much information about the relationship of that receptor with inflammation.

## 4. Glycine Transporters

There are several known families of glycine transporters, including solute-carrying transporter 36 (SLC36; also known as PATs), SLC38, and SLC6 (also known as GlyTs). Of these families, SLC6 is the only one specific to glycine. The SLC36 family members are expressed primarily in the intestine; the SLC38 family members are widely distributed throughout the body; and the SLC6 family members are expressed primarily in the intestine, kidney, and nervous system [71]. There are two known genes belonging to the SLC6 family, namely, GlyT1 and GlyT2. There are five variants of GlyT1 (a, b, c, d, and e) that differ at their N and C termini due to the use of alternative promoters and splicing, and there are three GlyT2 splice variants (a, b, and c) [37,72,73,74]. Both transporters use the energy stored in the transmembrane Na^+^ and Cl^−^ concentration gradients to transport glycine against a concentration gradient [75]. The specific stoichiometry is 2 Na^+^/1 Cl^−^/1 Gly for GlyT1 (SCL6A9) and 3 Na^+^/1 Cl^−^/1 Gly for GlyT2 [76]. These carriers have different mechanisms through which glycine is transported depending on the membrane voltage [77].

### 4.1. GlyT1

For GlyT1, most Na^+^ moves when glycine binds to the transporter, while for GlyT2, most Na^+^ charges move after glycine has dissociated into the cytosol (Figure 2) [77]. GlyT1 is expressed mainly in astrocytes, regulating the glycine concentration in the vicinity of NMDA receptors [78], and oocytes [79]. It is also found in the human intestine and is responsible for 30–50% of glycine uptake in intestinal epithelial cells, maintaining the glycine supply in enterocytes and colonocytes [80]. The kinetics of this transporter are characterized by the transition of the transporter attached to the substrate from the outside to the inside [77]. 

### 4.2. GlyT2

GlyT2 is located at the axon terminals of glycinergic neurons, mainly in the spinal cord and brain stem. These transporters facilitate glycine reuptake in presynaptic terminals [81]. The GlyT2-dependent influx of glycine allows for the maintenance of vesicular glycine reserves [82]. GlyT2 kinetics are governed by Na^+^ binding, which causes a conformational change [77].

## 5. Effects of Glycine in the Organism

### 5.1. Neurotransmission

Glycine participates in the rapid excitatory neurotransmission mediated by NMDA receptors. Indeed, the full activation of these receptors requires the binding of both glutamate and glycine [83,84]. This activity is necessary for plasticity processes such as learning, memory, and cognition [85,86]. These receptors are therapeutic targets in counteracting pathologies such as Alzheimer’s disease, Huntington’s disease, amyotrophic lateral sclerosis, schizophrenia, and other psychiatric diseases [87]. Glycine also modulates neurotransmission via GlyT1 and GlyT2 [88,89,90]. Mice lacking GlyT1 show severe respiratory and motor deficiencies due to hyperactive glycinergic signaling [91,92]. On the other hand, the participation of GPR6 due to glycine stimulation could decrease cAMP in striatal tissues and increase dopamine (improving movement) [93], showing a possible potential target for the treatment of Parkinson’s disease [94]. 

Glycine shows neuroprotective effects on neurons and microglia after ischemic stroke injury. It can inhibit nuclear factor kappa B (NF-κB) p65 and hypoxia-inducible factor 1α (Hif-1α) by activating AKT and downregulating phosphatase and tensin homologue (PTEN). This mechanism suppresses ischemia-induced M1 microglia polarization and inhibits inflammation [95]. Glycine confers protection against neuronal death in in vitro and in vivo experimental conditions [96], and clinical studies have shown that glycine can improve the prognosis of patients after ischemic stroke [97].

### 5.2. Antioxidant

Glycine effectively protects against alcohol-induced hepatotoxicity by reducing blood alcohol levels and the metabolic products of alcohol, reducing liver damage, and lowering the gastric emptying rate of ethanol [98]. Kupffer cells increase Ca^2+^ and release prostanoids and inflammatory cytokines in response to LPS stimuli, glycine acts by preventing increases in intracellular levels of Ca^2+^ in this kind of cell in concentrations comparable to serum levels in glycine-fed animals [99]. Furthermore, this amino acid plays a protective role against the toxic effects of oxidized oil since it has been observed that the co-administration of oxidized oils with glycine cause the recovery of liver structure and function, demonstrating glycine to be a beneficial amino acid that is protective against food toxicity [100].

### 5.3. Dietary Supplementation

In animal models, dietary supplementation with glycine reduces inflammation, morbidity, and mortality from pathogenic infections [101]. Betaine supplementation is associated with circulating levels of dimethylglycine. Betaine metabolism may be hampered by the activity of dimethylglycine dehydrogenase (DMGDH), which connects betaine to glycine synthesis via the demethylation of dimethylglycine into sarcosine [102]. Glycine supplementation has shown beneficial cardiac effects in a burn model [103] and in a skeletal muscle ischemia model, with decreased reperfusion-mediated necrosis and increased metabolic and functional recovery [104]. The protective effects of glycine have been reported for several sepsis models [105,106] and for hemorrhagic shock [107]. Normal levels of glycine in the body prevent lipopolysaccharide (LPS)-induced endotoxemia by binding to GlyRs on Kupffer cells [108]. Dietary supplementation with glycine decreases adipocyte size, adiposity, and the concentrations of free fatty acids and triglycerides in animal models [109,110]. It also inhibits the nonenzymatic glycation of proteins and hemoglobin in diabetic rats [111] and patients with T2DM [112,113].

### 5.4. Immunomodulation

Glycine exerts anti-inflammatory and immunomodulatory effects in several cell types [114,115]. However, the underlying mechanisms responsible for these beneficial effects remain unknown. There are some hypotheses about the actions of glycine in living organisms, and accumulating evidence suggests that glycine protects cells from oxidative stress-induced inflammation [116,117]. In a mouse model of cancer cachexia, glycine attenuated oxidative and inflammatory environments [118]. In a model of acute pancreatitis, glycine reduced the severity of pancreatic damage [119]. Moreover, glycine suppressed zymosan-induced joint inflammation [120]. Some studies have shown that supplementation with glycine can reduce acute and systemic allergic responses. Although the exact mechanism of action is still unknown, glycine may exert an effect on key effector cells, such as basophils and mast cells [117]. 

Glycine inhibits inflammatory responses and modulates the production of cytokines by many innate cells, including monocytes and intestinal and alveolar macrophages [121,122,123]. The best-studied glycine signaling pathway involves GlyRs, but glycine may also utilize GlyR-independent pathways to mediate its cytoprotective function, indicating that the presence of GlyR subunits is not a requirement for the anti-inflammatory effect of glycine [124].

Glycine is a potent therapeutic immunonutrient for various kinds of chronic liver conditions, including alcoholic liver disease [125]. Various insults in the CNS, such as ischemia, can promote neuroinflammation, with the polarization of microglia, the resident macrophages [126,127]. Treatment with glycine can attenuate neuroinflammation, suppressing M1 microglia polarization, the pro-inflammatory state [128], and promoting M2 polarization, the primary anti-inflammatory state [129]. Liu et al., 2019 [95], reported this benefit in an animal model of ischemic stroke: glycine antagonized ischemia-induced M1 polarization and promoted M2 polarization [95].

Glycine mitigates the adverse effects of LPS, peptidoglycans, and peroxisome proliferators. This amino acid modulates the secretion of cytokines and the activation of proteases and increased apoptosis in various animal models [130,131,132].

### 5.5. Low Glycine Plasma Levels Are Associated with Low-Grade Inflammation

Hyperglycemia or hyperinsulinemia reduce the average rate of glycine synthesis [7]. Based on previous studies, a low plasma glycine concentration is associated with hepatic insulin resistance, obesity, and T2DM [133]. This decrease is due to a reduction in glycine availability via the simultaneous participation of three distinct mechanisms: (a) decreased gut absorption, (b) decreased biosynthesis, and (c) increased catabolism or urine excretion [3]. 

Obesity and metabolic disorders show elevated plasma glucagon concentrations that reduce circulating glycine and increase its degradation [134,135]. On the other hand, an impaired hepatic branched-chain amino acid metabolism in obesity decreases circulating glycine concentrations [136]. Glycine supplementation (5 g/day or 0.1 g of glycine/kg/day for 14 days in association with N-acetylcysteine) improves the insulin response and glucose tolerance in patients with obesity [23,137].

## 6. The Role of Glycine in Inflammation

A glycine concentration of 1–3 mM triggers the opening of the GlyR ion channel [138], resulting in rapid hyperpolarization, a decrease in calcium, and a reduction in the synthesis of pro-inflammatory mediators, probably via the tumor necrosis factor α (TNF-α) receptor signaling pathway [51]. The GlyR-dependent pathway can modulate chronic and neuropathic pain [35,139,140] and the anti-inflammatory response (Figure 3) [114]. 

GlyRs are located on different cell types involved in immune responses, such as macrophages, monocytes, neutrophils, T lymphocytes [114,121,122], hepatic and alveolar macrophages [108], the pancreas, and Kupffer cells [135,141]. In macrophages, T lymphocytes, and neutrophils, GlyR activation suppresses the production of pro-inflammatory cytokines, thus supporting anti-inflammatory properties [121]. Glycine could modulate the low-grade inflammatory process through pathways that involve some of its targets that have already been identified in different cells. 

Glycine inhibits the cytokine synthesis stimulated by LPS, reduces serum transaminase levels, and decreases intracellular calcium concentrations by modulating chloride influx inside the cell [125,142]. In addition, there is evidence that glycine administration reduces the infiltration of inflammatory cells and attenuates the elevated concentrations of inflammatory cytokines and chemokines in the hepatic cells of mice via the inhibition of NF-κB [13]. In addition, glycine can upregulate adiponectin expression [143]. Glycine pre-treatment inhibits the inflammation of alveolar cells in LPS-induced lung injury [13,144]. Glycine bioactivity is anti-inflammatory in the lung and other tissues [12]. There is an inverse relationship between glycine plasma levels and systemic inflammation [135,145], and this effect is similar in obesity and diabetes [146]. 

Many researchers have attempted to describe the mechanism through which glycine exerts its beneficial effects. For decades, glycine has been proposed as an anti-inflammatory agent [114,121] and used as a therapeutic nutrient to treat inflammation related to diseases such as arthritis [132], gastric ulcers [147], melanoma [148], alcoholic liver disease [125], and endotoxic shock [107]. Glycine supplementation inhibits the expression of NF-κB through the inactivation of IκB, an upstream regulator of NF-κB signaling, as well as the production of its downstream pro-inflammatory cytokines and chemokines [13,149]. Glycine blocks the activation of NF-κB; hence, it promotes the downregulation of pro-inflammatory adipokines [107,150]. 

Glycine inhibits the production of pro-inflammatory cytokines like TNF-α, interleukin- 6 (IL-6), and IL-1β and increases the production of the anti-inflammatory cytokine IL-10 in activated macrophages, leucocytes [131], and T lymphocytes in a pathological state [151]. During endothelial inflammation, glycine can exert anti-inflammatory effects via the inhibition of the activation of NF-κB, degrading IκBα and increasing the expression of E-selectin and the production of IL-6 [152]. Glycine cytoprotection suppresses inflammation by preventing the immunogenic effects of necrosis and directly inhibiting pyroptosis. Moreover, efforts to understand the basis for glycine cytoprotection have led to the discovery of novel upstream immunomodulatory effects of glycine to block the primary activation of multiple types of inflammatory cells. This effect involves receptors that are different from those involved in glycine cytoprotection [124]. 

Glycine treatment attenuates the production of pro-inflammatory and Th2-skewing cytokines such as IL-13, TNF-α, and IL-4 in the RBL-2H3 basophilic cell line. The glycine effect on the cytokine response following IgE-mediated crosslinking on RBL-cells may be comparable to effects on mast cells [117]. Some studies have shown that the glycine-induced downregulation of NF-κB is mediated, at least partially, by NRF2 signaling [13,95]. The anti-inflammatory properties of glycine have been shown in animal models of septic shock [153]. Moreover, there is evidence that glycine treatment increases adiponectin and IL-10 messenger RNA (mRNA) expression [154].

Dietary supplementation with glycine has been proposed as a potential way to treat conditions with low-grade inflammation, such as obesity [155,156,157]. This approach has been proposed due to its ability to increase the mRNA expression of anti-inflammatory cytokines, such as adiponectin and IL-10 [123,154,158], and because glycine can inhibit the production of pro-inflammatory cytokines like TNF-α, IL-1β, and IL-6 [131,158]. In one study, 3 months of glycine treatment attenuated low-grade inflammation by decreasing pro-inflammatory cytokines [143]. Glycine decreases endotoxin-induced TNF-α production by Kupffer cells and alveolar macrophages and also reduces IL-1β and TNF-α expression while simultaneously stimulating IL-10 expression in LPS-activated monocytes [131]. It also interferes with TNF-α-mediated NF-κB activation in human coronary artery endothelial cells and differentiated 3T3-L1 adipocytes [152,156]. In animal models, supplementation with 0.5% glycine has been used to prevent the infiltration of inflammatory cells; to treat synovial hyperplasia in joints, edema, experimental arthritis, and lung inflammation [132,141]; and to inhibit tumor growth [130,148,159].

The pathophysiological mechanisms underlying glycine deficit and its potential clinical repercussions are not yet clear yet. It is important to mention that although glycine is a non-essential amino acid because it can be synthesized endogenously, glycine supplementation helps to maintain homeostasis (Figure 4).

It this review, we described the importance of glycine as an anti-inflammatory amino acid that not only has beneficial cosmetic effects but is also an alternative medicine for promoting homeostasis in an organism. It has been demonstrated that glycine can modulate the inflammatory response, using different targets in many cells in the entire organism. There are many studies that describe the indirect effects of glycine as a modulator of responses and mechanisms in the organism via dietary supplementation with glycine in animal models and in clinical trials. Another option is including the amino acid as a component of diet using animal models or in clinical trials, finally revealing the indirect role of glycine in reducing the inflammation process and decreasing some symptoms related to low-grade inflammation. However, there are few studies that have evaluated and followed changes due to glycine consumption in the diet over a long period of time, and those that have evaluated how glycine can decrease the risk or development of some diseases related to inflammation process or even the direct effects of glycine on cell activity and the response of cells to specific signals similar to an inflammatory environment.

## Figures and Tables

**Figure 1 ijms-24-11236-f001:**
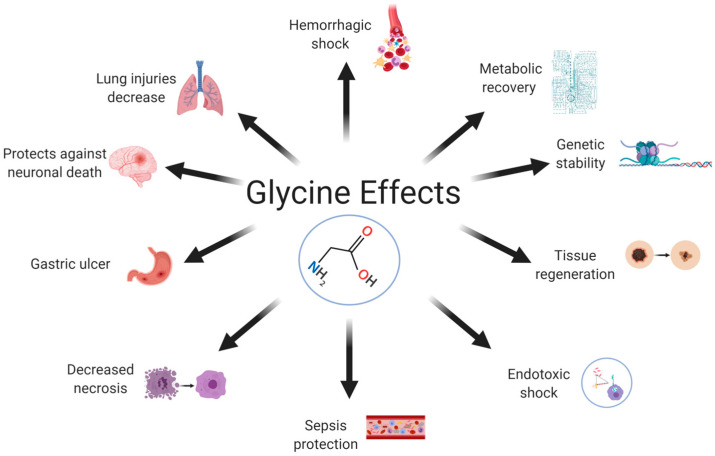
Glycine effects. Glycine is an aminoacid synthetized endogenous and there has been describe many activities in which it participates that include a variety of systems. Glycine has a protective effect in lung, brain, stomach, and intestine; participates in metabolic process; modulate process of the immune system such as tissue regeneration, decrease necrosis, sepsis protection; and finally, glycine is considerate as a genic expression modulator.

**Figure 2 ijms-24-11236-f002:**
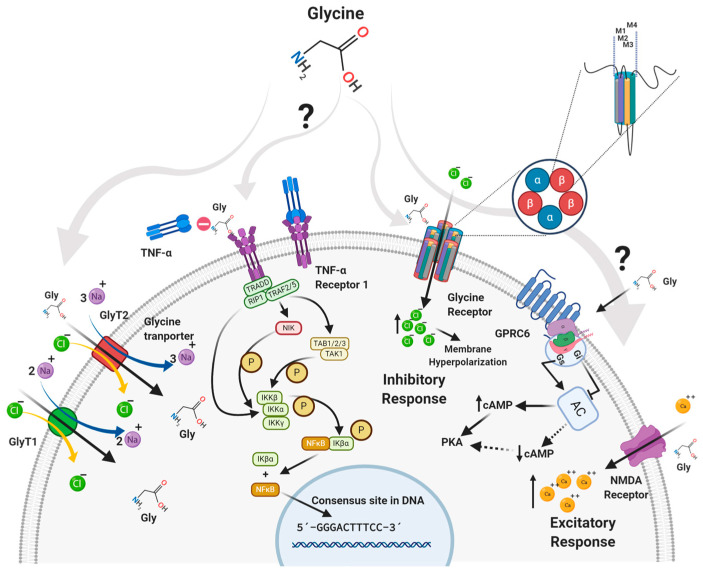
Glycine targets and pathways. According to the International Union of Basic & Clinical Pharmacology (IUPHAR) glycine has different targets such as Natural/Endogenous Targets: glycine receptor (consisting of glycine receptor α1, α2, α3, α4 and β subunits), ionotropic glutamate receptors co-agonist (GluN1, GluN2A, GluN2B, GluN2C and GluN2D) and GPRC6 Receptor. Transporters moving this compound across a lipid membrane with proton-coupled amino acid transporter 1, vesicular inhibitory amino acid transporter, Proton-coupled Amino acid Transporter 2, GlyT1, GlyT2, B0AT1, B0AT2, B0AT3, NTT4, sodium-coupled neutral amino acid transporter 1, sodium-coupled neutral amino acid transporter 2, sodium-coupled neutral amino acid transporter 4, Sodium-coupled neutral amino acid transporter 5 [27,28,29,30]. These may be the targets involved in the signaling by which glycine exerts its effect on the different cell lines of living organisms.

**Figure 3 ijms-24-11236-f003:**
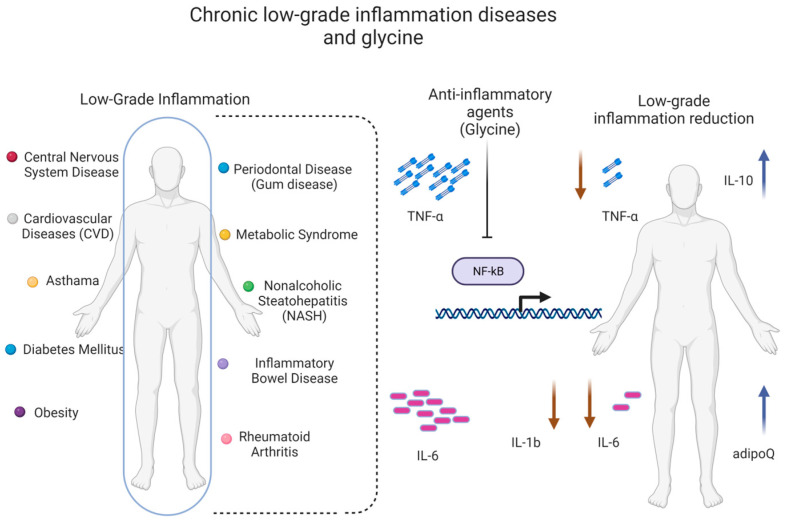
Chronic low-grade inflammation diseases and glycine. There are diseases that go through a low-grade inflammatory process and maintains decreased glycine concentrations in the body. Glycine, despite not being an essential amino acid, requires its consumption to maintain adequate concentrations in the body and to carry out its activity in the different organs. Glycine decreases the expression of proinflammatory cytokines through the inhibition of NF-kB and favors the expression of anti-inflammatory cytokines, thus reducing the feedback of the chronic inflammatory process that occurs in some diseases.

**Figure 4 ijms-24-11236-f004:**
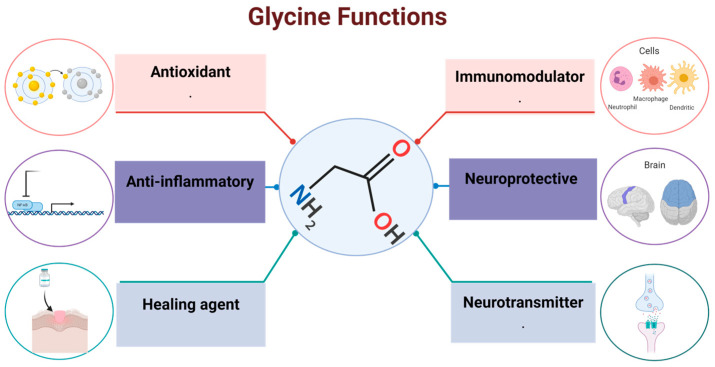
Glycine functions. Glycine function has been shown in different organs and tissues, from being a protein precursor to adjudicating functions such as antioxidant, immunomodulator, anti-inflammatory, healing agent, as well as, neuroprotective and neurotransmitter in central nervous system.

## Data Availability

Data Availability Statements are available in section “MDPI Research Data Policies” at https://www.mdpi.com/ethics.

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
