# Peer review of "Glycine: The Smallest Anti-Inflammatory Micronutrient"

_ijms, 2023, doi:10.3390/ijms241411236_

Round 1
Reviewer 1 Report (Previous Reviewer 4)
The manuscript has undergone some revisions by the authors, which have improved its content. As a result, it is now suitable for acceptance.
(If possible, please consider modifying the captions for the figures to provide more comprehensive explanations, as the current brief descriptions appear inadequate.)
Reviewer 2 Report (Previous Reviewer 1)
I have recommended acceptance of the manuscript in the previous round of review.
This manuscript is a resubmission of an earlier submission. The following is a list of the peer review reports and author responses from that submission.
Round 1
Reviewer 1 Report
The manuscript by Karla Aidee Aguayo-Cerón et al. integrates information from basic and clinical studies on the role of glycine as a therapeutic agent in regulating low-grade inflammation associated disease, and also discusses signal transduction pathways using glycine as ligands. It reviews the knowledge of glycine binding to specific receptors and transporters that are expressed in many types of cells throughout the tissues to perform their functions. The mechanism of glycine action is discussed on the basis of many existing studies, and the pathway related to the expression of NF-κB in many immune cells is highlighted as a potential mechanism for glycine to play its anti-inflammatory role, and the importance of dietary glycine supplementation to avoid the development of chronic inflammation is emphasized. However, the references are not up-to-date, and it lacks the summary of the most recent knowledge/publications, and does not conduct the corresponding discussion on the direction of further research, so the manuscript cannot be accepted in its present form.
Major issue:
1. As described in the abstract, the purpose of this review is to integrate information from basic and clinical studies on the role of glycine as a therapeutic agent to regulate low-grade inflammation associated with disease, i.e., the anti-inflammatory effects of glycine, but this content is not fully detailed in the main text.
2. Page 4 Line 146-148: It is mentioned that glycine has an effect on the expression of GPRC6, a receptor associated with neuroprotective activity, and there is not much information on its association with inflammation. The relationship between the GPRC6 receptor and inflammation is not clearly explained. What is the purpose of presenting this receptor here? It is suggested to supplement the importance of GPRC6 receptor in inflammation.
3. Page 5 Line 149: In the part of“4. Glycine Transporters”, it mainly introduces glycine transporters. It should introduce their practical application or importance in detail.
4. Page 6 Line 198: In the part of“5.2. Antioxidant”, the antioxidant properties of glycine and its practical application are not fully discussed.
5. It is suggested to supplement the summary and prospect of the anti-inflammatory effect of glycine at the end of this article.
Minor:
1. The full names of some abbreviations should be clearly displayed when they first appear in this manuscript. For example, GluN1, GluN2A, GluN2B, GluN2C, GluN2D, B0AT1, B0AT2, B0AT3, NTT4
2. Page 3 Line 114: “the channel opens”,Which channel, exactly?
3. Page 12 Line 513: The format of this reference is different from the others, please revise it to the same format.
4. Page 9 Line 341: There are few references in the past decade.
Author Response
Answer to reviewers and editorial team of the manuscript: “Glycine the smallest anti-inflammatory micronutrient ”
Manuscript ID: ijms-2306901
Reviewer 1
The overall idea of the study is very interesting and of importance for the scientific community. Nevertheless, there are some improvements to be made:
Reviewers' comments:
Comment
The manuscript by Karla Aidee Aguayo-Cerón et al. integrates information from basic and clinical studies on the role of glycine as a therapeutic agent in regulating low-grade inflammation associated disease, and also discusses signal transduction pathways using glycine as ligands. It reviews the knowledge of glycine binding to specific receptors and transporters that are expressed in many types of cells throughout the tissues to perform their functions. The mechanism of glycine action is discussed on the basis of many existing studies, and the pathway related to the expression of NF-κB in many immune cells is highlighted as a potential mechanism for glycine to play its anti-inflammatory role, and the importance of dietary glycine supplementation to avoid the development of chronic inflammation is emphasized. However, the references are not up-to-date, and it lacks the summary of the most recent knowledge/publications, and does not conduct the corresponding discussion on the direction of further research, so the manuscript cannot be accepted in its present form.
Major issue
- Comment
Thank you for your question, as described in the abstract, the purpose of this review is to integrate information from basic and clinical studies on the role of glycine as a therapeutic agent to regulate low-grade inflammation associated with disease, i.e., the anti-inflammatory effects of glycine, but this content is not fully detailed in the main text.
Answer
Thank you for your comment, the information was added to the manuscript
- Comment
Page 4 Line 146-148: It is mentioned that glycine has an effect on the expression of GPRC6, a receptor associated with neuroprotective activity, and there is not much information on its association with inflammation. The relationship between the GPRC6 receptor and inflammation is not clearly explained. What is the purpose of presenting this receptor here? It is suggested to supplement the importance of GPRC6 receptor in inflammation.
Answer
Thank you for your question, in that section we wrote about the targets of glycine and regarding to GPRC6 previous studies found changes in the expression of this receptor in cells treated with glycine, and according to an PCA (principal component analysis) that receptor cold be considerate as a marker with anti-inflammatory effects, but the information about the relation among GPRC6 and inflammation is low.
- Comment
Page 5 Line 149: In the part of “4. Glycine Transporters”, it mainly introduces glycine transporters. It should introduce their practical application or importance in detail.
Answer
Thank you for your comment,we include in the manuscript the information about the application of glycine transporters
- Comment
Page 6 Line 198: In the part of“5.2. Antioxidant”, the antioxidant properties of glycine and its practical application are not fully discussed.
Answer
Thank you for your question, we include in the manuscript the information about the antioxidant properties of glycine
- Comment
It is suggested to supplement the summary and prospect of the anti-inflammatory effect of glycine at the end of this article.
Answer
Thank you for your comment, we include in the manuscript the information according to the suggestion
Minor comments
- The full names of some abbreviations should be clearly displayed when they first appear in this manuscript. For example, GluN1, GluN2A, GluN2B, GluN2C, GluN2D, B0AT1, B0AT2, B0AT3, NTT4.
Answer
Thank you for your comment, GluN1, GluN2A, GluN2B, GluN2C, GluND, are subunits of glutamate receptor thata form the channel and does not have abbreviation
B0AT1, B0AT2, B0AT3, NTT4 the information was included in the text
- Page 3 Line 114: “the channel opens” , Which channel, exactly?
Answer
Thank you for your comment, The information was included in the manuscript line 114
- Page 12 Line 513: The format of this reference is different from the others, please revise it to the same format.
Answer
Thank you for your comment, the format of the reference was corrected according to the instruction
- Page 9 Line 341: There are few references in the past decade.
Answer
Thank you for your comment, the information was supplemented to the old references
Reviewer 2 Report
After reading the paper, this reviewer has several major concerns. 1, The mechanism by which glycine acts is not clear. So the authors cannot only focus on nuclear factor kappa B (NF-κB) . And this section is very short compared to the whole paper. 2, The title and subtitles in the paper should be rearranged. Can referee other review papers. 3, “5. Some Activities of Glycine” is a very strange section with different subtitles. What do the author mean?Author Response
Answer to reviewers and editorial team of the manuscript: “Glycine the smallest anti-inflammatory micronutrient ”
Manuscript ID: ijms-2306901
Reviewer 2
After reading the paper, this reviewer has several major concerns. 1, The mechanism by which glycine acts is not clear. So the authors cannot only focus on nuclear factor kappa B (NF-κB). And this section is very short compared to the whole paper. 2, The title and subtitles in the paper should be rearranged. Can referee other review papers. 3, “5. Some Activities of Glycine” is a very strange section with different subtitles. What do the author mean?
Answer
Thanks for reviewer comments:
- The main mechanism described for glycine is the pathway through the join of the amino acid to its receptor (GlyR), but there are other studies that find results in cell culture models and suggest that the amino acid can acts using different targets, in this review we focus in the mechanism related to the inflammatory process. In other hand, it has been described that NF-kB is a molecule that has an important role in the inflammation regulation. So, according to the objective of the review we decide to focus in this pathway to emphasis in the possible anti- inflammatory effect of glycine
- We consider the comments and made some changes in the subtitles.
- We change the subtitle “Some Activities of Glycine” instead of “Effects of glycine in the organism”
Reviewer 3 Report
The authors present a preview that focuses on the glycine, a non- essential amino acid as a dietary glycine supplementation to avoid the development of chronic inflammation. This review is definitely worth publishing. However, there a a few issues that needs to be highlighted. Please highlight the strength and limitation of this articles at the end. Also do include any clinical trials that has been carried out of the effect of glycine both in animals and humans and include how this data can be used to fill in the gap of knowledge that exists pertaining to glycine.
Author Response
Answer to reviewers and editorial team of the manuscript: “Glycine the smallest anti-inflammatory micronutrient ”
Manuscript ID: ijms-2306901
Reviewer 3
Comments
The authors present a preview that focuses on the glycine, a non-essential amino acid as a dietary glycine supplementation to avoid the development of chronic inflammation. This review is definitely worth publishing. However, there a a few issues that needs to be highlighted. Please highlight the strength and limitation of this articles at the end. Also do include any clinical trials that has been carried out of the effect of glycine both in animals and humans and include how this data can be used to fill in the gap of knowledge that exists pertaining to glycine.
Answer
Thanks for reviewer comments. We add some information at the end of the text. The aim of this review was to describe and propose a mechanism by which glycine acts and try to explain how the aminoacid decrease the inflammation process, and the information in which we base on are studies that use cellular or animal models and clinical trials, and include the references in the text .
Reviewer 4 Report
The manuscript entitled "Glycine the smallest anti-inflammatory micronutrient" describes it well, but it seems a copy-paste from somewhere. I strongly recommend the authors send back this manuscript along with a plagiarism report.
The review article “Glycine the smallest anti-inflammatory micronutrient”, is presented well, although there are serious deficiencies that need to be addressed properly in order to publish in IJMS.
1) The objective of the study should be clear both in the abstract and introduction portions.
2.) The whole review should be checked thoroughly and added with information, the present presentation is not quite enough.
3.) I didn’t see the summary and future perspectives of glycine in such study, it needs to be added properly.
4.) The figure’s caption is very poor, it needs to be resharpened well.
5.) Line 116, mem-brane should be corrected
6.) Line 120, glicine should be corrected.
7.) Line 215, put “.” After the reference number.
8.) Line 267 up to the last of the paragraph, organize it well, the GlyR location, all should be discussed like macrophages and others. The whole paragraph should be added with enough information.
Author Response
Answer to reviewers and editorial team of the manuscript: “Glycine the smallest anti-inflammatory micronutrient ”
Manuscript ID: ijms-2306901
Reviewer 4
The manuscript entitled "Glycine the smallest anti-inflammatory micronutrient" describes it well, but it seems a copy-paste from somewhere. I strongly recommend the authors send back this manuscript along with a plagiarism report.
Comments
1) The objective of the study should be clear both in the abstract and introduction portions.
Answer
Thanks for the comments, we made changes in the abstract and in the last paragraph of the section “Introduction”
2) The whole review should be checked thoroughly and added with information, the present presentation is not quite enough.
Answer
Thanks for the comment, we made changes in manuscript.
3.) I didn’t see the summary and future perspectives of glycine in such study, it needs to be added properly.
Answer
We included a paragraph at the end of the text to describe the future perspectives of glycine
4.) The figure’s caption is very poor, it needs to be resharpened well.
Answer
Thanks for the comment, we have modify the caption.
5.) Line 116, mem-brane should be corrected
Answer
The word was changed in the text
6.) Line 120, glicine should be corrected.
Answer
The word was changed in the text
7.) Line 215, put “.” After the reference number.
Answer
Thank for the comment, we have change in this section.
8.) Line 267 up to the last of the paragraph, organize it well, the GlyR location, all should be discussed like macrophages and others. The whole paragraph should be added with enough information.
Answer
Thanks for the comment, we have added more information in this section.
Round 2
Reviewer 1 Report
The authors have adequately addressed the questions raised in the 1st round of review, it's recommended for publishing without further revision.
Reviewer 4 Report
The revised manuscript submitted does not meet the necessary standards of quality. It needs further augmentation with additional information, better organization, and structure. Specifically, the summary, conclusion, and future perspectives must be presented as separate and distinct topics, with a clear and focused scope.